# High Frequencies of Genetic Variants in Patients with Atypical Femoral Fractures

**DOI:** 10.3390/ijms25042321

**Published:** 2024-02-15

**Authors:** Álvaro del Real, Raquel Cruz, Carolina Sañudo, José L. Pérez-Castrillón, María I. Pérez-Núñez, Jose M. Olmos, José L. Hernández, Carmen García-Ibarbia, Carmen Valero, Jose A. Riancho

**Affiliations:** 1Departamento de Medicina y Psiquiatría, Instituto de Investigación Sanitaria Valdecilla (IDIVAL), Facultad de Medicina, Universidad de Cantabria, 39011 Santander, Spain; delreala@unican.es (Á.d.R.); carolinasanudo@gmail.com (C.S.); olmosj@unican.es (J.M.O.); joseluis.hernandez@unican.es (J.L.H.); carmen.valero@unican.es (C.V.); 2Grupo de Medicina Xenómica, Centro de Investigación en Medicina Molecular y Enfermedades Crónicas, Universidade de Santiago de Compostela (USC), 15782 Santiago de Compostela, Spain; raquel.cruz@usc.es; 3Internal Medicine Department, University Hospital Rio Hortega of Valladolid, 47012 Valladolid, Spain; uvacastrv@gmail.com; 4Traumatology Department, University Hospital M. Valdecilla, 39008 Santander, Spain; isabel.perez@unican.es; 5Internal Medicine Department, Marqués de Valdecilla University Hospital, 39008 Santander, Spain; c.g.ibarbia@hotmail.es; 6Centro de Investigación Biomédica en Red de Enfermedades Raras (CIBERER), Instituto de Salud Carlos III, 28029 Madrid, Spain

**Keywords:** atypical femur fractures, genetics, SNPs

## Abstract

This study explores the genetic factors associated with atypical femoral fractures (AFF), rare fractures associated with prolonged anti-resorptive therapy. AFF are fragility fractures that typically appear in the subtrochanteric or diaphyseal regions of the femur. While some cases resemble fractures in rare genetic bone disorders, the exact cause remains unclear. This study investigates 457 genes related to skeletal homeostasis in 13 AFF patients by exome sequencing, comparing the results with osteoporotic patients (*n* = 27) and Iberian samples from the 1000 Genomes Project (*n* = 107). Only one AFF case carried a pathogenic variant in the gene set, specifically in the ALPL gene. The study then examined variant accumulation in the gene set, revealing significantly more variants in AFF patients than in osteoporotic patients without AFF (*p* = 3.7 × 10^−5^), particularly in ACAN, AKAP13, ARHGEF3, P4HB, PITX2, and SUCO genes, all of them related to osteogenesis. This suggests that variant accumulation in bone-related genes may contribute to AFF risk. The polygenic nature of AFF implies that a complex interplay of genetic factors determines the susceptibility to AFF, with ACAN, SUCO, AKAP13, ARHGEF3, PITX2, and P4HB as potential genetic risk factors. Larger studies are needed to confirm the utility of gene set analysis in identifying patients at high risk of AFF during anti-resorptive therapy.

## 1. Introduction

Atypical femoral fractures (AFFs) are low-trauma fractures that commonly occur in the subtrochanteric or diaphyseal region of the femur and have a horizontal or short oblique trajectory [1]. This type of fracture has been reported in some patients with monogenic bone diseases, such as hypophosphatasia, pycnodysostosis, osteopetrosis, X-linked hypophosphatemia (XLH), osteoporosis-pseudoglioma syndrome (OPPG), osteogenesis imperfecta (OI), and X-linked osteoporosis (OP). Apart from those rare cases, AFFs are notable for their association with prolonged bisphosphonate (BP) or other anti-resorptive therapy [2,3]. While the pathogenesis of AFFs remains unclear, the detection of these fractures in individuals who have not been exposed to BPs and in those with inherited bone disorders has raised the idea that genetic factors may contribute to the susceptibility of AFFs. In keeping with this concept, the incidence of AFFs shows large ethnic differences, being much more common among Asians. In the initial study by Pérez-Núñez et al., a connection between AFFs and several genetic variations was established via exon array analysis [4]. In addition, Roca-Ayats and colleagues detected 37 uncommon genetic alterations in the context of BP-associated AFFs, focusing on a trio of sisters. Among all these variations, two noteworthy genes stood out, GGPS1 and CYP1A1. GGPS1 plays a role in the mevalonate pathway, a significant contributor to cholesterol and steroidal hormone synthesis, which is also the target for amino-terminal BPs to inhibit osteoclast activity. CYP1A1 participates in steroid metabolism [5]. In recent years, various studies have tried to identify the genetic basis of AFFs [6,7]. However, the results have been inconclusive and largely not replicated.

Although genome-wide association studies (GWAS) have revealed numerous prevalent (usually noncoding) genetic variations linked to diseases, the utilization of whole-exome sequencing (WES) in association studies has emerged as a potent strategy for identifying drug targets. This is due to its enhanced ability to confidently identify effector genes, offering a clearer understanding of their directional impact [8]. Therefore, in this study, we used WES to pinpoint the genes contributing to the vulnerability of AFFs.

## 2. Results

After exome variant calling in AFF patients, in one case, we found a pathogenic variant in the ALPL gene, which encodes alkaline phosphatase. We later confirmed low serum levels of alkaline phosphatase in this patient. No pathogenic variants related to bone phenotypes were found in the remaining AFF cases.

Subsequently, the accumulation of variants, common or rare, was examined in the AFF and the osteoporosis groups. We observed a total of 31,537 ± 2579 variants among the 13 patients in the AFF group and 36,288 ± 301 variants among the 27 patients in the OP group. Since the sample size was small, we focused on a set of 457 bone-related genes (Appendix A). After filtering exome data by gene coordinates, implementing a full join with all samples, and removing variants that were present only in one group (to exclude possible differences in allele calling), we identified 1363 genetic variants that were present in at least one individual from each group. We then excluded 176 variants due to a departure from the Hardy–Weinberg equilibrium. Thus, the final analysis matrix had 1187 variants (Appendix A). The analysis at the individual variant level did not reveal statistically significant differences in the allele frequency distribution between the AFF and the OP control group (FDR > 0.05). 

However, in the pooled analysis combining the variants of the whole gene set, patients with AFFs tended to carry more variants than the osteoporotic ones (χ^2^ for linear trend *p*-value = 3.7 × 10^−5^) (Figure 1).

After grouping variants by gene, there were 12 genes showing significant differences in variant frequency (FDR < 0.05), comprising 120 variants (Table 1).

To replicate the variant distribution differences, we also compared the frequency of variants in the AFF group and the Iberian sample data (IBS) from the 1000 Genomes project (phase 3). Only 112 out of the 120 variants found in the genes pinpointed above were present in at least one individual of the IBS population. As shown in Table 2, the analysis of those 112 variants revealed significant differences between the AFF and IBS groups in seven genes (ACAN, AKAP13, ARHGEF3, P4HB, PITX2, SUCO, and UGT1A8).

## 3. Discussion

The present study aimed to investigate the genetic variants associated with individuals with AFFs, focusing on bone-related pathways. We obtained whole-exome sequencing data that were filtered to focus on a set of 457 bone-related genes. We found a pathogenic variant only in 1 out of 13 patients with AFFs. On the other hand, we found a higher variant accumulation in patients with AFFs compared with osteoporotic controls without AFF. 

The gene-level analysis was further replicated using the Iberian sample data from the 1000 Genomes Project. In particular, ACAN, AKAP13, ARHGEF3, P4HB, PITX2, and SUCO exhibited differential allele frequencies between the AFF group and both the OP control group and the Iberian population of 1000G. Overall, these results suggest that not only a complex interplay of genetic factors contributes to the susceptibility of osteoporosis [9], but also that most cases of AFFs are polygenic in nature. This finding is in line with our previous study using array genotyping, as well as with the inconsistent findings of other studies looking for pathogenic variants in patients with AFFs [3,4,6].

All six genes showing differential variant distribution are related to osteogenesis. SUCO (SUN domain-containing ossification factor) has been associated with skeletal dysplasia, osteopenia, and osteogenesis imperfecta [10]. Koide and collaborators found that Akap13 was expressed in bone tissue, and mice with haploinsufficiency of Akap13 (Akap13+/−) exhibited decreased bone mineral density, reduced bone volume/total volume ratio, decreased trabecular number, and increased trabecular spacing, mirroring the changes observed in the osteoporotic bone [11]. Moreover, P4HB missense mutations cause mild osteogenesis imperfecta [12], and also Cole–Carpenter syndrome [13,14]. Furthermore, through ARHGEF3 gene knockdown experiments and subsequent molecular analyses, relevant associations have been found between this gene and the expression of critical genes involved in bone metabolism. Notably, ARHGEF3 and its related gene RHOA appear to play roles as potential regulators of genes such as TNFRSF11B, ARHGDIA, PTH1R, and ACTA2, impacting both osteoblast-like and osteoclast-like cells [15]. In addition, PITX2 mutations are related to craniofacial and dental features of Axenfeld–Rieger syndrome patients [16]. ACAN gene (Aggrecan) is translated to a protein that is an essential component of the extracellular matrix of many tissues. The ACAN gene is particularly important in the formation and maintenance of cartilage [17]. Mutations in the ACAN gene lead to various skeletal disorders and conditions, including some forms of short stature and skeletal dysplasia. These conditions are often characterized by abnormal development of the bones and cartilage, leading to differences in height and bone structure [18,19].

Our study expands on the understanding of the genetic basis of AFFs, revealing novel insights into potential causal genes and pathways. Given that the variants analysed predominantly consist of common variants, our findings suggest that the genetic heterogeneity observed in AFFs may be influenced by polygenic factors. This emphasizes the complex interplay of genetic factors in determining susceptibility to AFFs, which may have implications for risk assessment and treatment strategies. The identification of significant genes across different analyses, as well as their validation in diverse populations, underscores the robustness of our findings. However, it is important to acknowledge the limitations of our study, including the relatively modest sample size and the need for further functional studies to elucidate the mechanistic underpinnings of the observed associations. We did not find significant between-group differences in the frequency of variant alleles at the individual variant level. Nevertheless, the power of the study was rather low for this type of analysis. The use of a group of patients with OP as the primary control for comparison may be criticized due to some epidemiological differences between that group and the AFF group, as well as the possibility that some patients in the OP group could develop AFFs in the future. However, the latter would bias our results toward the null hypothesis, thus reassuring us that the observed differences are not false positives. Additionally, we were interested in the genetic factors leading to AFFs, not in the factors leading to osteoporosis. Thus, a group with osteoporosis, rather than a healthy people group, seemed preferable for the primary comparison with the AFF group.

Our analysis revealed a predominance of single nucleotide polymorphisms (SNPs). These variants encompassed both conservative and non-conservative changes. Notably, the variants were distributed across exons and untranslated regions that could be implicated in binding domains as well as signalling domains, suggesting potential diverse functional consequences on protein–protein interactions and intracellular signalling cascades crucial for bone metabolism regulation. Overall, besides the pathogenic ALPL variant found in one patient, only 22–26 variants were annotated as VUS/pathogenic/likely pathogenic in ClinVar or InterVar (see Appendix A). Most variants were common, relatively frequent variants. Therefore, most of them were predicted to be benign/likely benign, as expected.

In conclusion, our study provides compelling evidence for the genetic heterogeneity of AFF. Our findings suggest that the accumulation of variants in genes such as ACAN, AKAP13, ARHGEF3, P4HB, PITX2, and SUCO contribute to determining the risk of AFFs in patients on anti-resorptive drugs. The individual susceptibility to AFFs seems to be determined by polygenic factors in most cases, as well as drug therapy and other acquired influences. In fact, only a minor proportion of patients carry single pathogenic gene variants. Further studies on these genes could be informative in determining which patients are at higher risk of developing AFFs when treated with anti-resorptive agents. Given the polygenic origin suggested by the present and other studies, perhaps the use of polygenic risk scores would be worthwhile.

## 4. Materials and Methods

### 4.1. Patients

Thirteen patients with AFFs were included. AFFs were diagnosed according to published criteria [1]. Most of them were included in a previous genotyping study [4]. As a comparison group, we used a group of patients with early-onset osteoporosis (*n* = 27) who had not experienced AFFs. Patients with FFA included 13 women with a mean age of 75+/−12 years. All of them had been treated with antiresorptives (bisphosphonates, 12; denosumab, 1). Patients with osteoporosis without FFA (*n* = 27), of 64+/−7 years of age; 16 had received anti-resorptive drugs (bisphosphonates, 15; denosumab, 1). The study protocol was approved by the institutional review board (Comité de Ética en Investigación Clínica de Cantabria), and all patients gave informed written consent.

### 4.2. Whole Exome Sequencing

DNA was extracted from blood leukocytes using commercial methods. After quality control and quantitation, samples were sequenced in the Fundación de Medicina Genómica (Santiago de Compostela, Spain) using the Illumina NovaSeq 6000 platform, employing KAPA HyperExome probes (Roche) to capture the regions of interest, which included exons and flanking intron regions. All samples had read depth coverage ≥30× for 96% of the targeted regions, with a coverage average of 114. The human_g1k_v37 was used as the reference genome. Data alignment, analysis, and variant call were conducted using the following computational tools: DRAGEN-OS version 0.2020.08.19, SAMBLASTER version 0.1.26, GATK (Genome Analysis Toolkit) version 4.4.0.0, Pindel version 0.2.5b9, Picard version 3.0.0, mosdepth version 0.3.3, bedtools version 2.31.0, samtools version 1.17, ExomeDepth version 1.1.16, Haplogrep version 2.4.0, SnpEff, and ANNOVAR.

### 4.3. Gene Set Selection

After the search for pathogenic variants through the whole exome data, we explored the prevalence of common and rare variants in a set of 457 genes related to skeletal homeostasis. Those genes were selected by an educated search of published data, including the PanelApp list of osteogenesis imperfecta [20]; the genes described in AFF reviews [2,7]; and genes from the mevalonate pathway, as used in [21]. 

### 4.4. Downloading of 1000G Data

All variants annotated by samples of the phase 3 analysis were downloaded from the ftp site of 1000 Genomes, in the GRCh37 version. Samples of the IBS population and the loci analysed were filtered by using BCFtools [22]. Variants appearing in the subset of 1000 Genomes and in our subset of samples were merged and analysed with R software (version 4.3.0).

### 4.5. Data Analysis and Statistics

Data filtering and statistical analyses were conducted using R software (version 4.3.0). The matrices of allele distribution in the various groups were compared by the individual variant level, gene level, and gene set level. Fisher exact tests were used to compute the *p*-values, and then the false discovery rate (FDR) values were estimated to control the type II error associated with multiple tests [23]. Thus, differences with an FDR < 0.05 were regarded as statistically significant.

## Figures and Tables

**Figure 1 ijms-25-02321-f001:**
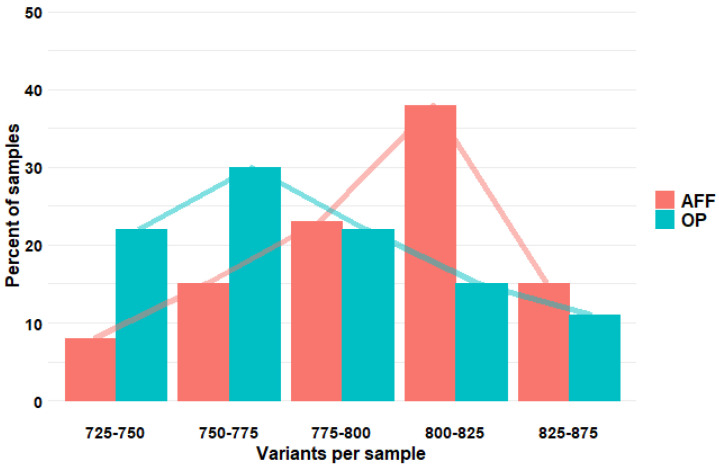
The number of variants accumulated in the bone-related gene set in patients belonging to the OP group (green) or the AFF group (red). The X-axis shows the number of variants in each individual (grouped in intervals), and the Y-axis represents the number of individuals (as a percentage within each group).

**Table 1 ijms-25-02321-t001:** Gene-level analysis. The number of variants in each gene that were present in at least one individual in each group. Frequency of the alternative alleles, and statistical significance of between-group comparisons, both at the allele level and the genotype level.

GENE	Nº Variants	Alternative Allele Frequency (%)		FDR (*p*-Value)	
		Control OP	AFF	Allele	Genotype
ACAN	16	47	59	9. × 10^−3^	2.85 × 10^−2^
AKAP13	21	41	34	2.10 × 10^−1^	2.11 × 10^−10^
APC	7	61	68	8.88 × 10^−1^	3.48 × 10^−2^
ARHGEF3	6	29	37	8.84 × 10^−1^	3.44 × 10^−2^
CYP2D6	10	20	27	4.20 × 10^−1^	1.29 × 10^−5^
NBN	3	20	44	1.48 × 10^−2^	2.66 × 10^−3^
NOTCH2	7	23	32	4.89 × 10^−1^	3.44 × 10^−2^
P4HB	3	19	26	1.00	2.91 × 10^−2^
PITX2	2	8	35	9.82 × 10^−3^	2.85 × 10^−2^
SPP1	3	37	44	1.00	2.85 × 10^−2^
SUCO	5	23	42	9.82 × 10^−3^	3.31 × 10^−2^
UGT1A8	37	27	27	1.00	2.85 × 10^−2^

**Table 2 ijms-25-02321-t002:** Gene-level comparisons of variant allele distributions in AFF patients and IBS 1000 Genomes population.

GENE	Nº Variants	Alternative Allele Frequency (%)		FDR (*p*-Value)	
		Control IBS	AFF	Allele	Genotype
ACAN	16	53	59	4.93 × 10^−2^	3.47 × 10^−2^
AKAP13	21	47	34	6.08 × 10^−8^	5.62 × 10^−23^
ARHGEF3	6	25	37	1.19 × 10^−2^	8.88 × 10^−3^
P4HB	3	14	26	2.21 × 10^−2^	8.38 × 10^−3^
PITX2	2	18	35	2.21 × 10^−2^	3.47 × 10^−2^
SUCO	5	29	42	1.19 × 10^−2^	8.88 × 10^−3^
UGT1A8	34	23	28	1.19 × 10^−2^	8.88 × 10^−3^

## Data Availability

Data is contained within the article and Appendix A.

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
