# Peer review of "High Frequencies of Genetic Variants in Patients with Atypical Femoral Fractures"

_ijms, 2024, doi:10.3390/ijms25042321_

Round 1
Reviewer 1 Report
Comments and Suggestions for Authors
Review Comments
The authors conducted an interesting study using WES to find the plausible associations of structural variants with AFF pathogenesis. The Introduction is well-written, also, methods and Discussion. However, this is an interesting work, it needs minor revision.
#Comment 1: Please reconsider about your title; it is better to change into this: High frequencies of genetic variants in patients with atypical femoral fractures
#Comment 2: The authors did not mention about the variants themselves in any part of the paper. A table for variants is needed with the columns containing chromosome location, Ref allele, Altered allele, Variant type [function: missense, missense+regulatory, etc. ], HGVS, RS ID, and Gene Symbol. This should be added to the results section.
#Comment 3: The authors should mention about the ACMG classification of each variant and discuss about the ClinVar report(s) for each variant. The ACMG classification define the level of pathogenicity of the variants; for example, some variants might be Pathogenic, or Likely Pathogenic or even a strong VUS. The authors can add a column in the Comment 1 table entitled “ACMG classification”. They would be able to widen the genetic standpoint of the paper. This classification will help the paper to prioritize the variants, consequently, the level of paper will increase and more special.
#Comment 4: Discuss about the variants and their functions, this can be described by the residue changes, INDEL, stop-gained, stop-lost, etc. Check the variant locations for finding any regulatory region that a variant may play role. This will help to make questions about the transcriptomic roles of these variants in AFF.
Comment 5: In discussion section, the authors emphasized on the pathways of genes involved in the AFF pathogenicity, but did not perform any signaling-based analysis; thus, it is highly recommended that the authors should carry out a bioinformatics-based analysis for their suggested genes. This will increase the reliability and validity of their findings. There are well-known applications such as Cytoscape and Enrichr. The authors will be able to discuss about the signaling-based outputs and suggest valuable strengthening clues or suggesting future studies to investigate the possible gaps.
Reviewer 2 Report
Comments and Suggestions for Authors
This study describes a whole exome sequencing analysis of the variants contributing to atypical femoral fractures in anti-resorptive therapy. It concludes that a heterogeneity in the accumulation of gene variants that regulate osteogenesis, rather than pathogenic variants, present a risk for susceptibility to atypical femoral fractures when compared to osteoporosis patients and a general population database.
The study is well-presented and easy to follow for someone with limited experience in genomic approaches. One minor point of interest:
1. The discussion of the osteogenic genes on page 4 is good. For biologists, could the authors offer a general comment on any trends on the nature of the variants within the osteogenic gene groups?
Are the variations SNPs, insertions/deletions, alternative splicing, etc?
Are variants conservative or non-conservative changes from the control
groups?
How might variants affect the protein function, such as accumulated variants
skewed toward exons specifying an extracellular binding domain versus a
signaling domain?
This reviewer acknowledges that a detailed description of different gene variants
might be outside the scope of this manuscript.
